



# High-resolution stratospheric volcanic SO$_2$ injections in WACCM

Emma Axebrink[1], Moa K. Sporre[1], and Johan Friberg[1]

[1]Department of Physics, Lund University, Lund 22100 Sweden

**Correspondence:** Moa K. Sporre (moa.sporre@fysik.lu.se)

**Abstract.** Aerosols from volcanic eruptions impact our climate by influencing the Earth's radiative balance. The degree of their climate impact is determined by the location and injection altitude of the volcanic SO$_2$. To investigate the importance of utilizing correct injection altitudes we ran climate simulations of the June 2009 Sarychev eruptions with three SO2 datasets, in the Community Earth System Model Version 2 (CESM2) Whole Atmosphere Community Climate Model Version 6 (WACCM6).

We have compared simulations with WACCM's default 1 km vertically resolved dataset M16 with our two 200 m vertically resolved datasets, S21-3D and S21-1D. The S21-3D is distributed over a large area (30 latitudes and 120 longitudes), whereas S21-1D releases all SO$_2$ in one latitude and longitude grid-box, mimicking the default dataset M16.

For S21-1D and S21-3D, 95% of the SO$_2$ was injected into the stratosphere, whereas M16 injected only 75% to the stratosphere. This difference is due to the different vertical distribution and resolution of SO$_2$ in the datasets. The larger portion of

SO$_2$ injected into the stratosphere for the S21 datasets leads to more than twice as high sulfate aerosol load in the stratosphere for the S21-3D simulation compared to the M16 simulation during more than 8 months. The temporal evolution in AOD from two of our simulations, S21-3D and S21-1D, follows the observations from the space-borne lidar instrument CALIOP closely, while the AOD in the M16 simulation is substantially lower. This indicates that the injection altitude and vertical resolution of the injected volcanic SO$_2$ substantially impact the model's ability to correctly simulate the climate impact from volcanic

eruptions.

The S21-3D dataset with the high vertical and horizontal resolution resulted in global volcanic forcing of -0.24 W/m$^2$ during the first year after the eruptions, compared with only -0.11 W/m$^2$ for M16. Hence, our study high-lights the importance of using high-vertically resolved SO$_2$ data in simulations of volcanic climate impact, and calls for a re-evaluation of further volcanic eruptions.




# 1   Introduction

Aerosols impact our climate by influencing the Earth's radiative balance. Directly by scattering and absorbing solar radiation or indirectly via impact on cloud properties. These effects result in a net cooling effect on the climate. Aerosol emissions from fossil fuel combustion have counteracted some of the warming effects of anthropogenic greenhouse gases. However, aerosols'
climate impact is still a subject of great uncertainty (IPCC, 2021). It is important to understand natural sources of aerosols in order to better understand how humans affect the climate via emissions of greenhouse gases (Myhre et al., 2013; Robock, 2000).

Explosive volcanic eruptions that inject effluents into the stratosphere are a natural source of $SO_2$, which can have a large impact on the climate (Robock, 2000). The volcanic $SO_2$ is converted into sulfuric acid aerosol forming particulate matter,
which can create years of negative radiative forcing by scattering incoming solar radiation (Sigl et al., 2015). The aerosol is eventually removed from the stratosphere in the extratropics when the air is transported to the troposphere (Sigl et al., 2015; Gettelman et al., 2011; Appenzeller et al., 1996; Solomon et al., 2011). The severity of the climate impact is determined by the explosiveness of the eruption, the $SO_2$ mass, the injection altitude, and the location of the volcano (Robock, 2000; Kremser et al., 2016).

Volcanic eruptions have from time to time substantially cooled the Earth's climate (Sigl et al., 2015). The 1991 Mt. Pinatubo eruption is the latest eruption where a large amount of $SO_2$ reached high up into the atmosphere and lowered the global averaged surface temperature by several tenths of a degree Celsius (Kremser et al., 2016). Apart from such large size eruptions, less explosive eruptions have also proved to have a significant effect on the climate (Andersson et al., 2015; Vernier et al., 2011; Friberg et al., 2018). Volcanic cooling by multiple smaller volcanic eruptions caused significant cooling in the end of the
1990s and beginning of the 2000s (Santer et al., 2015; Andersson et al., 2015).

The vertical distribution of $SO_2$ from a volcanic eruption is crucial information, since the altitude determines the residence time of the aerosols (Andersson et al., 2015; Friberg et al., 2018; Kremser et al., 2016; Robock, 2000). Aerosols in the stratosphere can have a residence time of several years whereas tropospheric aerosols have a residence time of weeks or less (Kremser et al., 2016). Stratospheric aerosols thus have a prolonged climate impact compared to tropospheric aerosols (SPA, 2006). For
a volcanic eruption to affect the climate more long-term, the emitted sulfur needs to reach the stratosphere, i.e. be an explosive volcanic eruption. Less explosive eruptions often position the $SO_2$ in the vicinity of the tropopause. To estimate the climate impact of such eruptions, it is of particular importance to place the $SO_2$ at the correct altitude (Schmidt et al., 2018).

To investigate volcanic eruptions and their climate impact, global Earth System Models (ESMs) can be utilized. Global modelers often use satellite observations of volcanic $SO_2$ to estimate the climate impact of volcanic eruptions. Most $SO_2$ satellite
instruments are passive sensors and therefore lack direct vertical measurements. The altitude of the $SO_2$ clouds are therefore indirectly estimated resulting in coarse vertical resolution with substantial uncertainties. Clarisse et al. (2014) showed that IASI can provide $SO_2$ data with vertical resolution down to ∼2 km, and MIPAS has a vertical resolution of 3-5 km (Höpfner et al., 2015). This is on the order of one magnitude coarser than typical $SO_2$ layers from the June 2009 Sarychev eruptions (Sandvik et al., 2021). In Sandvik et al. (2021) we combined passive satellite measurements from the AIRS (Atmospheric Infrared





Sounder) satellite instrument with the active satellite sensor CALIOP (Cloud-Aerosol Lidar with Orthogonal Polarization) and created an $SO_2$ inventory with approximately 60 meter vertical resolution. With this method we create a 3D dataset where we provide altitude information for different $SO_2$ layers from the same eruption emitted at different times and altitudes.

ESM simulations of explosive volcanic eruptions' climate impact are generally run with $SO_2$ released as a column in the grid-boxes above, or in the vicinity of, the location of the volcano (Timmreck et al., 2018). This requires that the meteorology

and tropopause height are simulated correctly in order to represent the transport of the volcanic aerosol during the first few days after the eruption. Small errors in horizontal or vertical transport can cause large errors in the evolution of the $SO_2$ distribution, leading to inaccurate simulations of aerosol formation, and ultimately its resulting climate impact. Using a 3D dataset retrieved a few days after the eruption could reduce such uncertainties.

To investigate the importance of utilizing a high vertically and horizontally resolved volcanic $SO_2$ emission dataset, we have

implemented the dataset of Sandvik et al. (2021) in an ESM. We have modeled the eruptions of Sarychev Peak in June 2009. This volcano is located in the Northern Hemisphere (NH) at the center of the Kuril islands (48.092°N 153.20°E). This case is considered to be a complex series of volcanic eruptions since it erupted for several days and injected $SO_2$ over a wide range of altitudes. The duration of the eruption was from the 11th to the 16th of June, spreading $SO_2$ from 11-19 km altitude. The total mass of $SO_2$ emitted from the eruptions has been reported to range from 0.6 to $1.2 \pm 0.1$ Tg (Carboni et al., 2016; Haywood

et al., 2010).

In this study, we ran three simulations with different $SO_2$ emission datasets with the Community Earth System Model version 2 (CESM2.1), Whole Atmosphere Community Climate Model (WACCM6). The first is WACCM's default volcanic $SO_2$ column dataset with an assumed vertical profile, at 1 km resolution. The second is a dataset at 200 m vertical resolution where the $SO_2$ is distributed over a wide geographical region representing the initial spread of $SO_2$ based on Sandvik et al.

(2021). The third dataset is a hybrid between the first two and constitutes a column dataset at 200 m vertical resolution compiled from Sandvik et al. (2021). All simulations are evaluated by comparison to aerosol observations from the satellite sensor CALIOP.



## 2   Method

In this section, we describe the SO$_2$ datasets used in the global Earth system model, how they were created, and the differences
between them. A brief model description is also included in this section and a description of the satellite dataset we compare
the model simulations to.

### 2.1   SO$_2$ data

We have implemented the SO$_2$ dataset of the 2009 Sarychev Peak eruption described in Sandvik et al. (2021). It was compiled
by combining horizontally resolved SO$_2$ data from the satellite-instrument Atmospheric Infrared Sounder (AIRS) aboard the
satellite Aqua, with the vertical aerosol profiles from the satellite-instrument CALIOP. The SO$_2$ and aerosol observed from
these instruments were assumed to be co-located and therefore have the same height profile. The aerosol data from CALIOP
(at 60 m resolution) was coupled to the SO$_2$ data from AIRS using the dispersion model FLEXPART (FLEXible PARTicle
dispersion model), enabling retrieval of vertical profiles of the SO$_2$ layers with a high resolution (Sandvik et al., 2021).

The Sarychev Peak erupted multiple times over several days, starting on the 11 of June and continuing for 5 days. However,
most of the SO$_2$ was emitted on the 15 of June (Rybin et al., 2011). The dataset from Sandvik et al. (2021) contains data from
AIRS swaths around midnight between the 18 and 19 of June. The (Sandvik et al., 2021) 3D dataset has a vertical resolution
of 1 K in potential temperature, $61\pm 56$ m, or $1.8\pm 2.9$ mbar. In this study, we regridded this dataset to a vertical resolution
of 200 m and a horizontal resolution of $0.95°$ latitude $\times$ $1.25°$ longitude during the implementation into WACCM.

### 2.2   Model description

Simulations were run with the Specified Dynamic (SD) version of the WACCM6 (Gettelman et al., 2019). WACCM6 is an
extension of the Community Atmosphere Model version 6 (CAM6), and part of the Community Earth System Model Version
2 (CESM2.1) (Danabasoglu et al., 2020). WACCM6 is a global high-top atmospheric model, spanning from the surface to the
thermosphere. WACCM6-SD has a top altitude of 140 km with 88 levels and we have ran the model with a horizontal resolution
of $0.95°$ latitude $\times$ $1.25°$ longitude. We have run the model with an active atmosphere and land, but prescribed sea-surface
temperatures (SSTs) and sea-ice concentrations (Gettelman et al., 2019).
WACCM6 includes advanced atmospheric chemistry in the troposphere, stratosphere, mesosphere, and lower thermosphere
(TSMLT). The chemistry includes 231 solution species, and the following chemical reactions; 150 photolysis reactions, 403
gas-phase reactions, 13 tropospheric, and 17 stratospheric heterogeneous reactions. For the stratospheric reactions, three types
of aerosol particles are included, sulfate, nitric acid trihydrate, and water-ice (Gettelman et al., 2019). Sulfates in the strato-
sphere are produced by the chemical oxidation of SO$_2$ by the OH radical. The sulfate will then, via intermediate steps, produce
H$_2$SO$_4$ gas (Liu et al., 2012; Mills et al., 2017). The H$_2$SO$_4$ gas can either condensate on excising or form new particles
through binary H$_2$SO$_4$-H$_2$O nucleation (Vehkamäki et al., 2002, 2013). The newly formed particles are added to the Aitken
mode after growth according to the parameterization from Kerminen and Kulmala (2002).



**Table 1.** Properties for the three datasets of $SO_2$.

| Simulation name | S21-3D | S21-1D | M16 |
|---|---|---|---|
| Vertical resolution | 200m | 200m | 1km |
| Horizontal resolution | $0.95° \times 1.25°$ | Single-column | Single-column |
| Vertical distribution | 11 - 19 km | 11 - 19 km | 11 - 15 km |
| Release date | 19th of June | 15 - 16th of June | 15 - 16th of June |
| $SO_2$ | 1.09 Tg | 1.09 Tg | 1.2 Tg |

WACCM6 utilizes the Modal Aerosol Model, four-mode version, (MAM4) as standard. This includes Aitken, Accumulation,
Coarse, and a Primary carbon mode (Liu et al., 2016). MAM4 in WACCM6 includes modifications of the aerosol code to
better represent aerosol processes in the stratosphere (Mills et al., 2016). The MAM4 gas-aerosol exchange module treats
stratospheric sulfate as aqueous $SO_4^=$. The $H_2SO_4$ equilibrium vapor pressure treats condensation and evaporation of $H_2SO_4$
in the stratosphere to allow for shrinkage and growth between the Accumulation and coarse mode (Mills et al., 2016).

The Specified Dynamic (SD) version (WACCM6-SD) allows the simulations to be nudged. We have nudged with Modern-
Era Retrospective analysis for Research and Applications, version 2 (MERRA2) from the surface to 50 km with a relaxation
between 50 and 60 km and no nudging above 60 km. The horizontal winds and surface pressure were nudged while temperature
nudging was not used.

## 2.3  Simulation description

Three different simulations, referred to as S21-3D, S21-1D, and M16, were run over the period of January 2009 to December
2010 to investigate the eruption of Sarychev Peak in 2009 with different vertical and horizontal resolutions of $SO_2$ datasets as
input. The differences between the datasets are summed up in Table 1 with further details below.

The first simulation, M16, contains the default $SO_2$ dataset, Volcanic Emissions for Earth System Models, version 3.11
(Neely and Schmidt, 2016, VolcanEESM), for the Sarychev eruption from WACCM6. For 2009 and 2010, all eruptions except
Sarychev's were removed. M16 is a single column (1D) emission dataset with a vertical resolution of 1 km. 0.6 Tg of $SO_2$
was released on two occasions, 15 and 16 of June, e.g. a total of 1.2 Tg. The $SO_2$ was released over a time period of 6 hours,
starting at 12:00 UTC and ending at 18:00 UTC (Neely and Schmidt, 2016; Mills et al., 2016).

The second simulation, S21-3D, contains a volcanic $SO_2$ dataset for the Sarychev eruption and was created from the work
of Sandvik et al. (2021). This dataset has a vertical resolution of 200 m and a horizontal resolution of $0.95°$ latitude $\times$ $1.25°$
longitude. The $SO_2$ is vertically distributed between 10 and 19 km and horizontally between the longitudes $130°E$ and $130°W$,
Figure 1. The S21-3D dataset releases all 1.09 Tg $SO_2$ over a time period of two hours, starting on the 19th of June at 00:30
UTC and ending at 02:30 UTC.





The third simulation, S21-1D, utilizes the dataset of the first simulation but with the horizontal distribution summed up, making the dataset a single column (1D) emission file. The dataset has the same vertical resolution of 200 m as the S21-3D dataset. The $SO_2$ is released on the 15 and 16 of June over a time period of 6 hours, starting at 12:00 UTC and ending at 18:00 UTC. The total amount released is the same as for S21-3D, 1.09 Tg. This dataset was created to mimic the M16 dataset, described above, in all aspects apart from having a higher vertical resolution.

The first five months of the simulations were run without any volcanic forcing and served as a spin-up. The three simulations, S21-3D, S21-1D, and M16, were run as branches from the spin-up simulation for an additional 19 months, from the first of June 2009 to the last of December 2010. We have also ran a simulation without any volcanic emissions (No-Volc).

The differences in the vertical and horizontal profile for the three $SO_2$ emission datasets are shown in Fig. 1. S21-3D and S21-1D have identical vertical profiles as shown in Fig. 1a. We can clearly see that much of the $SO_2$ in the S21-3D and S21-1D is located at a higher altitude compared to the default dataset M16. S21-3D and S21-1D are also more spread vertically compared with M16. Figure 1b shows the horizontal distribution of the $SO_2$ input dataset in simulation S21-3D. The red triangle marks the location for Sarychev Peak and is the location where M16 and S21-1D release the $SO_2$. The several eruptions from the Sarychev peak during these days reached different altitudes, leading to the broad horizontal distribution seen in Fig 1 b-d. The $SO_2$ layers located around 140°W was injected at higher altitude and has the majority of the $SO_2$ mass located at around 15 km. The $SO_2$ layers located around 130°E is positioned at lower altitudes with the majority of the mass at approximately 12-13 km altitude. The Eastern and Western $SO_2$ layers were transported in very different directions relative to the volcano clearly displaying the complexity of this eruption.

## 2.4 Aerosol data - satellite-derived aerosol extinction coefficients

The model simulations were compared with aerosol extinction data compiled from satellite observations retrieved by the spaceborne lidar CALIOP. The sensor acquired data at 532 and 1064 nm, and had a polarization filter to retrieve depolarization data at 532 nm. We used nighttime data in the latest version of the lowest level available, i.e. the Level 1b v4-51 (Product CAL_LID_L1-Standard-V4-51 ). Data were screened for ice clouds in the lowest 3 km of the stratosphere using depolarization ratios, and polar stratospheric cloud data were removed using a temperature threshold of 195 K outside 60°S - 60°N (see Friberg et al. (2018), Martinsson et al. (2022) and Friberg et al. (2023) for details). Backscattering coefficients were computed by correcting for light attenuation by particles and molecules (including ozone) throughout the stratosphere (Friberg et al., 2018; Martinsson et al., 2022; Friberg et al., 2023). Extinction coefficients were computed using a lidar ratio of 50 sr, i.e. a typical extinction to backscattering value for volcanic aerosol (Jäger and Deshler, 2002, 2003).



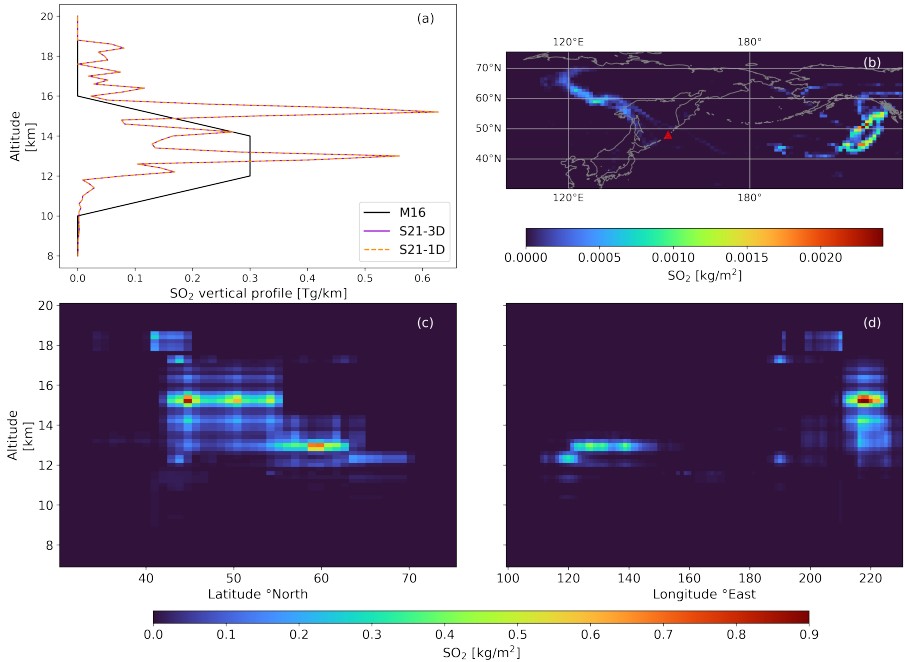

**Figure 1. (a)** Vertical SO$_2$ profiles for the three input datasets of each simulation. The vertical profile for M16 and S21-1D is the summed total injection for the eruption on the 15th and the 16th of June, whereas the vertical profile for S21-3D is the total injection on the 19th of June. **(b)** Horizontal resolution of the SO$_2$ dataset on the 19th of June for the S21-3D simulation. The red triangle marks the location of the volcano Sarychev Peak. **(c)** distribution of SO$_2$ over latitude and altitude. **(d)** distribution of SO$_2$ over longitude and altitude

## 3 Results and discussion

### 3.1 Temporal and spatial evolution of volcanic SO$_2$

The injected volcanic SO$_2$ profiles in the three simulations result in a large difference in SO$_2$ lifetime. Figure 2 shows the increase in global SO$_2$ load in the atmosphere following the June 2009 eruptions of the Sarychev peak. The volcanic SO$_2$ from M16 and S21-1D was injected on the 15 and the 16 of June with a total of 1.2 Tg for the M16 and 1.09 Tg for the S21-1D dataset. The S21-3D injected the SO$_2$ on the 19 of June with a total mass of 1.09 Tg. SO$_2$ disappears more quickly for the M16 simulation (Fig. 2 black line) compared to the simulations with S21-1D and S21-3D (orange and purple lines), regardless of the 0.11 Tg larger injected SO$_2$ mass in M16. The more rapid removal occurs since a large fraction of SO$_2$ in M16 is injected at altitudes below the tropopause, where the SO$_2$ is subject to the rapid wet chemistry of the troposphere, causing the SO$_2$ to be removed more quickly compared to the S21-1D and S21-3D datasets (Fig. 3). In the S21-1D and S21-3D simulations, more than 95% of the total SO$_2$ mass was injected into the stratosphere whereas only 75% of the SO$_2$ was injected into the stratosphere in the M16 simulation.





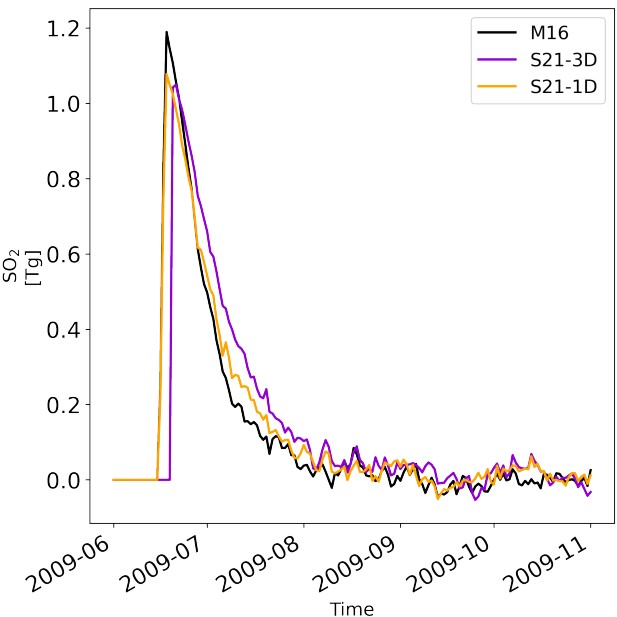

**Figure 2.** Global evolution of volcanic SO$_2$ in the M16, S21-1D and S21-3D simulations. To isolate the volcanic SO$_2$ we have subtracted the SO$_2$ levels in the No-Volc simulation from the other 3 simulations.

The time evolution of the vertical distribution of the SO$_2$ concentration is shown in Fig. 3. The volcanic SO$_2$ is seen at 6 different times, 5 (a), 12 (b), 19 (c), 26 (d), 33 (e), and 40 (f) days after the volcanic eruption on the 15 of June. Both
the stratospheric SO$_2$ mass (solid lines) and the total atmospheric (tropospheric + stratospheric) SO$_2$ mass (dashed lines) are shown. A large fraction of the SO$_2$ mass at lower altitudes are located in the troposphere in the M16 simulation. This is seen in Fig.3a where the dashed line deviates from the stratospheric mass (solid line). The tropospheric SO$_2$ is removed rapidly, shown by the difference between the dashed and solid line for the M16 simulation, where most tropospheric SO$_2$ was removed already 12 days after the eruption (Fig 3b). There is very little difference between the solid and dashed lines for the S21 simulations
demonstrating that most of this SO$_2$ is injected into the stratosphere. Not only is a larger fraction of SO$_2$ in the S21 simulations located in the stratosphere, the stratospheric SO$_2$ is also located at a higher altitudes, i.e. deeper into the stratosphere. This leads to higher SO$_2$ concentrations in the S21 simulations, in particular between 100 and 200 hPa. Also the horizontal SO$_2$ distribution impact the lifetime of the SO$_2$. In M16, SO$_2$ is spread more towards the subtropics, where the tropopause is located at high altitudes, likely leading to more rapid cross-tropopause transport, reducing the stratospheric SO$_2$ mass (Fig. S1).
Even though the vertical SO$_2$ profiles for the two S21 datasets are rather similar after 5 days there is a pronounced difference in the maximum SO$_2$ concentrations up to one month after the simulation (Fig. 3). The difference between the two S21 simulations is most likely a result of differences in horizontal spread of the SO$_2$ in the two simulations, where SO$_2$ in S21-1D is transported more towards the subtropics leading to more cross-tropopause transport for S21-1D than S21-3D. This exemplifies the importance of models' ability to realistically simulate the air movement and weather patterns at the time of the volcanic





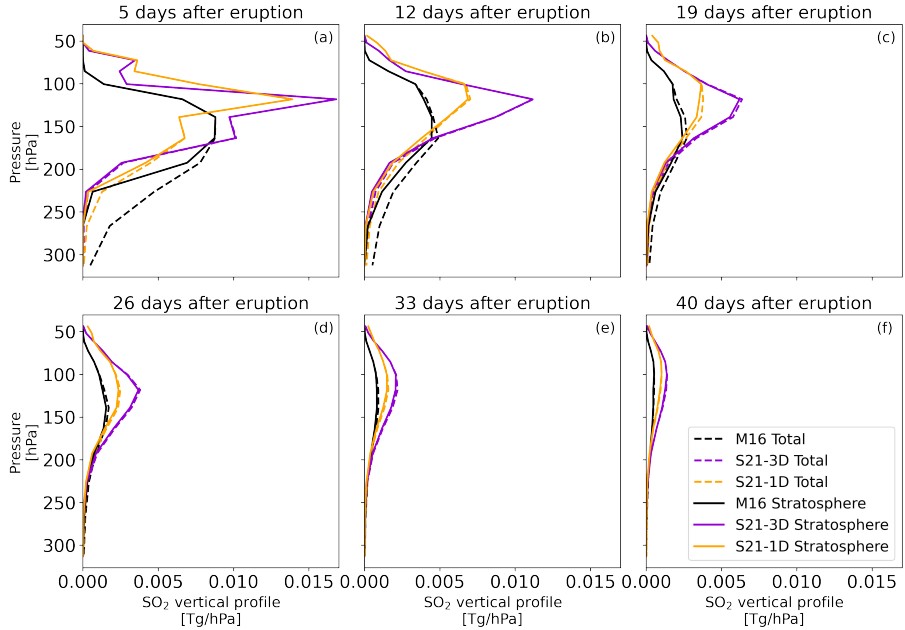

**Figure 3.** Vertical profile for volcanic $SO_2$ at 5 (**a**), 12 (**b**), 19 (**c**), 26 (**d**), 33 (**e**), and 40 (**f**) days after the volcanic eruption on the 15 of June. The dashed lines represent the total amount of $SO_2$ in the atmosphere whereas the solid lines represent the total amount of $SO_2$ in the stratosphere. To isolate the volcanic $SO_2$ we have subtracted the $SO_2$ levels in the No-Volc simulation from the other 3 simulations.

eruption. Simulations of volcanic climate impact are often run with column data of $SO_2$, where the volcanic injections are implementing vertical columns in single geographical (latitude × longitude) grid cell. Small errors/uncertainties in simulated air dynamics can result in vast differences in the geographical spread of the volcanic $SO_2$, leading to under- or overestimation of the aerosol lifetime and resulting climate cooling. Using the S21-3D dataset from satellite observations a few days after the eruption, where the initial transport has already taken place, reduces the importance of the models' ability to correctly simulate the air movement at the time of the eruption.

### 3.2 Temporal and spatial evolution of volcanic $SO_4$

The injected $SO_2$ is converted to $SO_4$ over the first months after the injection. Figure 4 shows the resulting increase of $SO_4$ after the volcanic eruption together with the decreasing $SO_2$ in the stratosphere. The peak mass for $SO_4$ differs both in time and magnitude for the three simulations. In the M16 simulation $SO_4$ peaks in mid July, four weeks after the eruption. The S21-1D and S21-3D volcanic $SO_4$ peaks in August, approximately 8 weeks after the eruption.

The earlier peak date for M16 than S21-1D and S21-3D stems from the difference in their vertical profiles of $SO_2$, where S21-1D and S21-3D injected more $SO_2$ to higher altitudes. In M16, a larger fraction of the $SO_2$ is injected into the first few kilometers above the tropopause. Both the injected $SO_2$ and the resulting aerosol formed at these lower altitudes are transported out of the stratosphere more quickly than $SO_2$ and aerosol located at the higher altitudes, explaining the longer-lasting $SO_4$





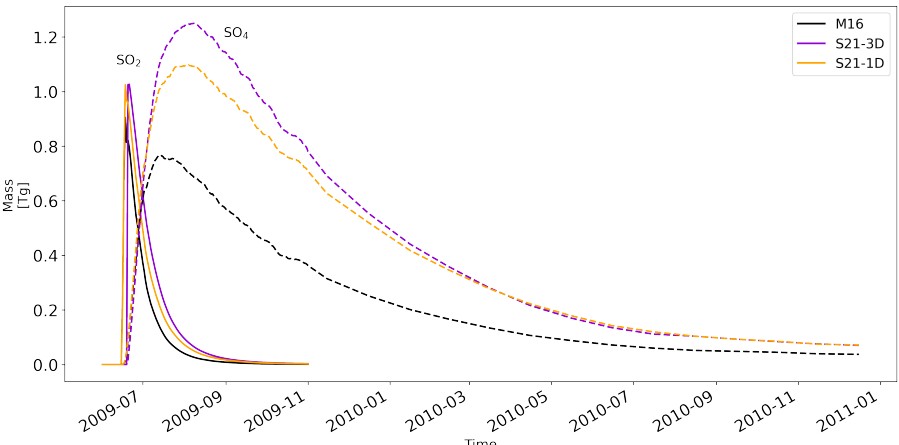

**Figure 4.** Stratospheric evolution of $SO_2$ (gas) and $SO_4$ (particle phase) over time. Daily values for both $SO_2$ and $SO_4$ til the end of October 2009, monthly values for $SO_4$ from November 2009 to December 2010. To isolate the volcanic $SO_2$ and $SO_4$ we have subtracted the $SO_2$ levels in the No-Volc simulation from the other 3 simulations.

and later peak for S21-1D and S21-3D. The $SO_4$ mass for S21-3D is substantially larger than for M16 already in July and remains higher throughout fall. In November the $SO_4$ mass is almost twice as high for S21-3D compared with M16 indicating a substantially larger volcanic climate impact in the S21-3D simulation. The $SO_4$ mass one and a half years after the eruption, December 2010, is still elevated for all three simulations. The S21 datasets have however an almost double amount of $SO_4$ mass at the end of 2010.

The large differences in volcanic sulfate aerosol loading over time is also visible in Fig. 5 The initial transport of the volcanic $SO_2$ results in different patterns in the $SO_4$ load between the datasets emitted as a column and the S21-3D dataset. After this, the pattern of the $SO_4$ load is similar between the simulations but aerosol concentrations drop of more rapidly in the M16 simulation compared to the S21 datasets. The aerosol is mainly located at mid and high latitudes for all three simulations but there is substantial equatorward transport during the NH autumn and winter after the eruption.

### 3.3 Comparison with CALIOP observations

Here we will compare the simulations with aerosol observations from the space-borne lidar CALIOP. This comparison is done for the aerosol extinction coefficient (Fig. 6) and AOD (Fig. 7). The first four columns in Fig. 6 represent simulations with our three datasets, M16, S21-1D, S21-3D, and CALIOP observations, where each row corresponds to monthly zonal mean values from June to November 2009. The fifth column in the figure shows the average aerosol extinction over all longitudes, i.e. extinction profiles. Since CALIOP is a polar orbiting satellite and only nighttime data from CALIOP is used in this study, there is missing data at high latitudes in the NH, in particular during the summer months. We have removed the data from the missing latitudes for all simulations to enable a direct comparison. We have also introduced a common tropopause mask to ensure that we compare data from the same latitudes and altitudes. All model simulations initially show lower extinction values in the




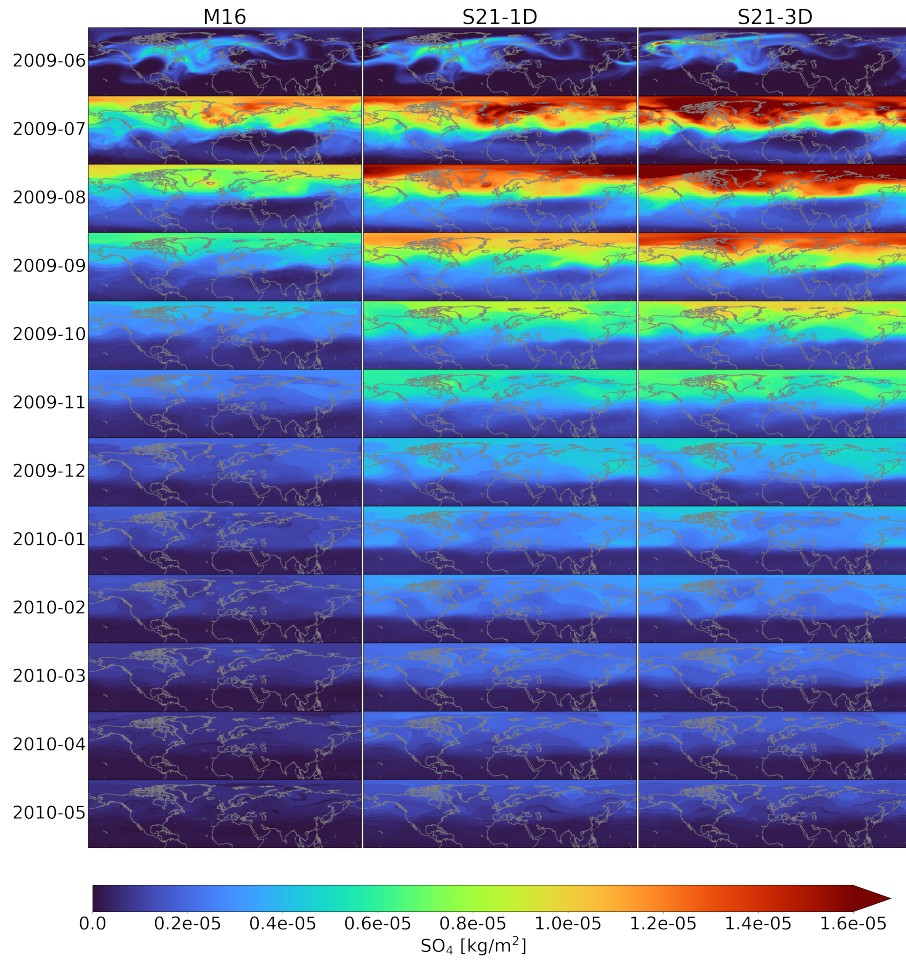

**Figure 5.** Monthly mean of stratospheric $SO_4$ in the NH during the first year post the volcanic eruption. To isolate the volcanic $SO_4$ we have subtracted the $SO_2$ levels in the No-Volc simulation from the other three simulations.

lowermost troposphere than the CALIOP observations. Averaging data in the proximity of the tropopause is complicated due

to the strong concentration gradients in this altitude region. The satellite data contain a substantially higher vertical resolution of both the extinction data and tropopause altitude than the models do. The coarser resolution of the model results in less sharp concentration gradients in the tropopause region. Moreover, for the simulations, the division between the stratospheric and tropospheric data was done based on the maximum probability of the daily chemical tropopause which results in that some of the lowest stratospheric data include influence from tropospheric air which will lower the extinction values. Above these

lowest altitudes, S21-1D and S21-3D both show more similar extinction coefficients compared with the CALIOP observations.

There are clear differences in the altitude-latitude distributions among the three simulations, where the S21 simulations show higher extinction coefficients in the northern midlatitude LMS. Aerosol, in all simulations, spread to the tropics, but not to as high altitudes in the M16 as in the S21 simulations. This is expected due to the generally lower injection altitudes for the



**Table 2.** Global average volcanic effective radiative forcing for the 3 simulations for different time periods.

| Volcanic ERF | 2009 | 2010 | June 2009 - May 2010 |
|:---:|:---:|:---:|:---:|
| M16 | -0.11 | -0.018 | -0.11 |
| S21-3D | -0.19 | -0.092 | -0.24 |
| S21-1D | -0.16 | -0.061 | -0.20 |

simulations with the M16 $SO_2$ dataset. The simulations predict lower extinction coefficients at the lowest kilometers of the
235 northern midlatitudes and larger volcanic influence at higher altitudes. CALIOP shows the highest extinction coefficients at
low altitudes, which is expected due to the higher pressure there. Furthermore, CALIOP shows that almost all aerosol remained
below 20 km altitude. Thus, it did not reach the upper branch of the BD circulation. Even though there are some differences
between the three simulations and the CALIOP observations the general patterns are however similar; The Sarychev eruption
i) influenced mainly the midlatitudes, ii) was almost isolated within the NH, and iii) did not enter the deep BD branch.

The extinction coefficients for the simulations and observations start to attain similar values and gradients at most altitudes
in August, following the initial phase of $SO_2$ transformation and particle formation (Jun-Jul), with the M16 showing the
lowest extinction coefficients. The S21 simulations continue to agree with observations in the following two months, whereas
M16 starts to deviate more from the observations and show lower extinction coefficients than both observations and the S21
simulations. This pattern is most pronounced in the LMS, illustrating the influence of outflow from the stratosphere which
leads to the lower AODs for M16 than for the S21 observations.

### 3.4 Radiative forcing - comparison of simulations

Finally, we will evaluate the extent of volcanic climate cooling estimated by the three simulations. Figure 8 shows the global
clear sky volcanic effective radiative forcing for the simulations. The effective radiative forcing (ERF) was calculated using
the method from (Ghan, 2013; Schmidt et al., 2018). The S21-3D simulation, run with $SO_2$ at high vertical and horizontal
resolution, predicts the highest and longest impact on the global volcanic forcing. The datasets with only high vertical resolution
but released in one column, S21-1D, follow the curve of S21-3D closely but with slightly lower values. The dataset with low
vertical resolution M16, has the weakest global clear sky volcanic forcing which disappears more rapidly compared to the other
two simulations. The peak value for the M16 simulation is -0.36 $W/m^2$ in August, the peak value for S21-1D is -0.41 $W/m^2$
in July and the peak value for S21-3D is -0.52 $W/m^2$ in August. The long-term forcing differed more among the models. The
255 forcing during the first year post eruption was more than twice as high for the S21-3D than for simulations with the models
default dataset M16, i.e. -0.24 and -0.11 $W/m^2$, respectively (Table 2). This large difference exemplifies the importance of
implementing the volcanic $SO_2$ injections at at correct altitudes in models.

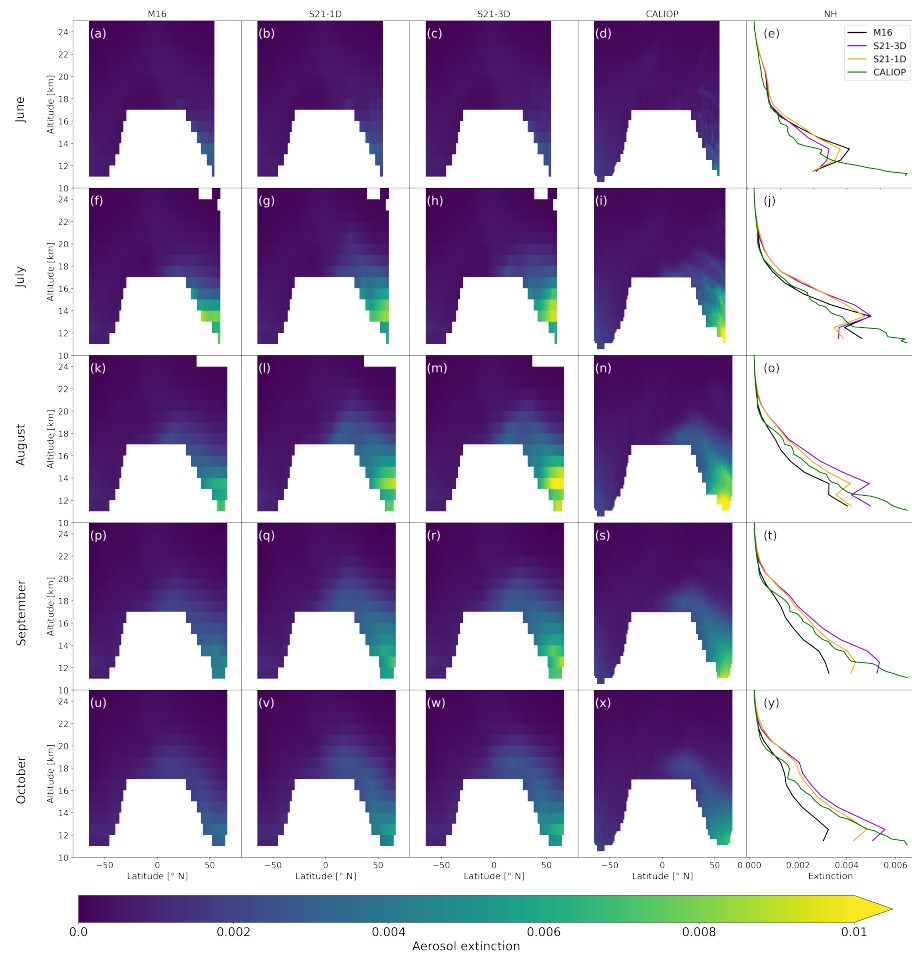

**Figure 6.** Monthly mean stratospheric evolution of the aerosol extinction coefficient for the three simulations and satellite observations from CALIOP. The first three columns show the simulations, (M16, S21-1D, and S21-3D), and the fourth column represents the CALIOP observations. The fifth column shows the average vertical aerosol extinction profiles in the NH for both simulations and the observations. The rows correspond to different months, starting from June to November 2009. The white areas are excluded values located in the troposphere, and missing latitudes in CALIOP. Note that the simulations have a wavelength of 550 nm whereas CALIOP observations have a wavelength of 532 nm.




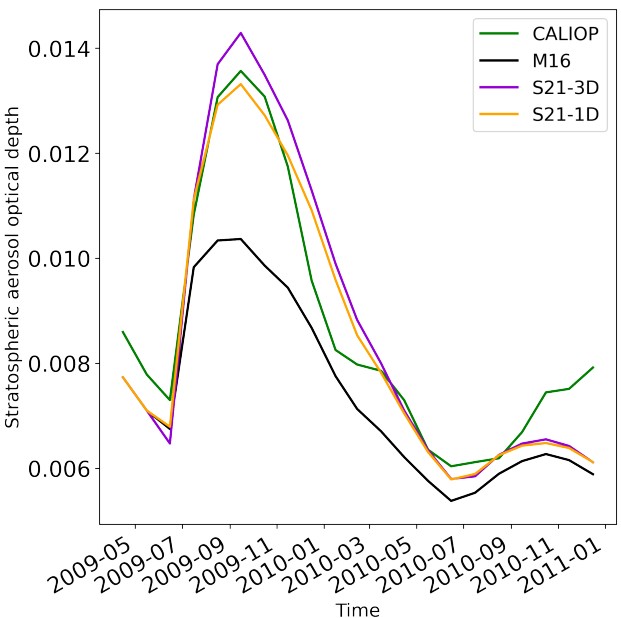

**Figure 7.** Stratospheric Aerosol optical depth (AOD) for the three simulations M16, S21-3D, and S21-1D, compared with observations by CALIOP. Note that the simulations show AOD at 550 nm whereas CALIOP observations provide AODs at a wavelength of 532 nm.

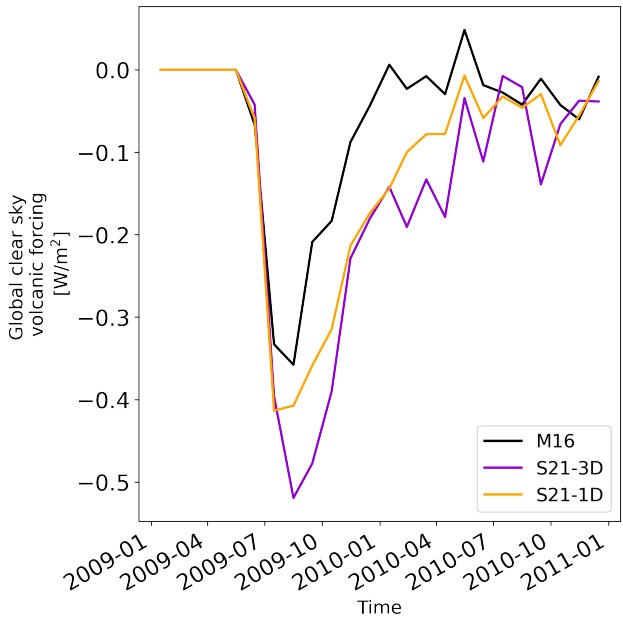

**Figure 8.** Global clear sky volcanic forcing from the Sarychev eruption for the three model simulations.



## 4 Conclusions

We have simulated the Sarychev eruptions' impact on the stratosphere and climate, using three different $SO_2$ injection profiles
in WACCM (Whole Atmosphere Community Climate Model). The eruptions positioned $SO_2$ throughout the lowest strato-
sphere and upper troposphere, in an altitude range of 11-19 km, increasing the stratospheric aerosol load (AOD) by 100% in
the months following the $SO_2$ injection. The overarching goal of this work was to investigate the influence of vertical $SO_2$ dis-
tributions on the stratospheric aerosol load and climate. To this end, we compared our simulations with high-vertical resolution
observations from the satellite borne lidar instrument CALIOP.

WACCM simulations with high-resolution $SO_2$ data captured the AOD well in the aftermath of the June 2009 Sarychev
eruptions. Simulations with these datasets produced very similar temporal evolution in stratospheric AODs as observations
from the satellite borne high-vertical resolution lidar instrument CALIOP. Furthermore, the simulated vertical distribution of
the aerosol load, expressed by the aerosol extinction coefficients, agreed well with the CALIOP observations. On the other
hand, simulations with the default volcanic injection dataset showed generally lower aerosol extinction coefficients and AODs,
and could not reproduce the observed values.

Simulations with high-resolution $SO_2$ data produced more than twice as strong volcanic forcing as the default dataset in
WACCM. The global clear sky radiative forcing during the first year after eruption amounted to -0.24 (-0.11) W/m$^2$ for the
high (low) resolution dataset. Although, holding 10% more $SO_2$, the default dataset induces far less climate cooling than
the high resolution datasets do. These findings highlight the need to produce high-vertical resolution datasets of volcanic
$SO_2$ injections to the stratosphere and indicate that our present understanding of volcanic climate cooling is limited by the
$SO_2$ profiles. Furthermore, it is highly likely that not only the Sarychev eruption's climate cooling is underestimated due to
inaccurate assumptions on $SO_2$ profiles. Climate cooling of pre- and post-Sarychev eruptions may to varying degrees be under-
or overestimated due to limited knowledge of the $SO_2$ vertical profiles. This highlights the need for further investigations of
volcanic $SO_2$ profiles. Our study required high-vertical resolution satellite retrievals of aerosols which have until present only
been accomplished by lidar. CALIOP provided us with such data from 2006 - 2023. This study highlights the usefulness of
spaceborne lidar systems, the need for continuous atmospheric observations from such systems, and exemplifies the need for
future space borne lidars.

*Code and data availability.* CESM is an open source model that is available to download through git, instructions are found here:
https://www.cesm.ucar.edu/models/cesm2/download. The $SO_2$ input files for all simulations are available here: 10.5281/zenodo.11192344.
Also monthly averaged model output from the simulations and monthly averaged CALIOP are available trough this link. CALIOP lidar data
are open-access products available via https://search.earthdata.nasa.gov/search?fp=CALIPSO.



*Author contributions.* E.A. performed the model simulations with WACCM. E.A. did most of the data analysis with contributions from M.K.S and J.F.. J.F. compiled the aerosol extinction coefficient data from CALIOP. E.A. wrote the majority of the paper. M.K.S and J.F. wrote parts of the paper. All authors have contributed to the discussions regarding the manuscript.

*Competing interests.* The authors declare that they have no conflict of interest.

*Acknowledgements.* The computations and data handling were enabled by resources provided by the National Academic Infrastructure for Supercomputing in Sweden (NAISS) and the Swedish National Infrastructure for Computing (SNIC) at Tetralith (project no 2023/22-1104, 2023/6-311, 2023/1-13 and 2024/23-95) partially funded by the Swedish Research Council through grant agreements no. 2022-06725 and no. 2018-05973. The CALIOP Level 1b lidar data were produced by NASA Langley Research Center.



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
