# Peer review of "Impact of $SO_2$ injection profiles on simulated volcanic forcing for the Sarychev 2009 eruptions - investigating the importance of using high vertical resolution methods when compiling $SO_2$ data"

_EGUsphere, 2024_

## Referee Comment (RC1)

Review on:
**High-resolution stratospheric volcanic SO2 injections in WACCM**
Emma Axebrink et al.
egusphere-2024-1448
June 7, 2024

The paper by Axebrink et al. highlights an important aspect for simulations of the sulfate aerosol evolution in the stratosphere after a volcanic eruption. They performed simulations with three different SO2 emission profiles and show that the results depend strongly on the profile used.

Emission rate, altitude, profile and timing are important for a realistic simulation. The evolution of the sulfate aerosol formation is very non-linear and depends on SO2 and background concentrations. The emission altitude determines the direction in which the volcanic cloud is transported in the model. Many of these aspects are nicely highlighted in the paper.

The paper is well written and reads well. I recommend publication after a few minor corrections.

**General:**

The article misses the aspect of microphysical processes and the consequent differences in particle size between the different simulations. Scattering of sulfate aerosols depends on the particle size (e.g. Laakso et al 2022). This could be another reason for the differences in the simulated forcing. My recommendation is to take a look at the particle size, e.g. calculate the effective radii, and include a short discussion.

The description of your experiments is a bit misleading. The table implies that your vertical grid of the S21s simulations has a vertical grid spacing of 200 m. This is not the case. Instead, you must interpolate the input data onto the model grid. Right? So it is not the resolution of the injection input data that is important. What is more important is the injection profile in the model. What does it look like and what does the model grid look like? Figure 6 shows quite similar profiles for June. So I assume that the height resolution of the input data disappears in the model. You can show and explain more details and show an initial phase profile.

A clear reason for the differences in the forcing calculations is the simulated altitude of the volcanic cloud (Fig 6). Surprising is the different behaviour between S21-1D and S21-3D. A better explanation should be added.

Have a look for other papers of volcanic eruptions, especially using WACCM, and add some citations.

**Specific comments:**

Line 38: What means significant here?

Line 43: A residence time of several years is quite unrealistic.

Line 49: No, the climate impact is estimated from the simulations. Satellites provide estimates on eruption rate and altitude.

Line 62: ....climate impact. Do you have a reference?

Line 63: Maybe, but due to the non-linear nature of the aerosol formation your simulations miss the early phase of the eruption. The may lead to different particle sizes and aerosol concentration.

Line 72-73: Add a reference for the dataset.

2.2 Model description: Information on the vertical grid are important here.

Table 1: Very misleading. Starting with 'Simulation name' and 'Vertical resolution' this could also be the vertical resolution of the model. What means single column? Should be in the model 0.95 x 1.25 as well.

Fig 1: b) shows the vertical integral, the burden, of SO2?

Line 165: SO2 does not disappear.

Line 184 and line 187-188: This argument is valid if your volcanic cloud stays at the same altitude (in meters). But from Fig 6 you can see a different behavior of the cloud. So the sulfur evolution cannot be deduced from Fig 1S alone. Microphysical processes are also important. You may have different particle size, different sedimentation, and different altitude, all of which affect the lifetime.

Line 189: The wording is difficult here. In principle, a model simulates one out of many possible weather/climate conditions. So, a model has its own reality. As you nudge the model, the results should be close to reality. On the other hand, a strongly nudged model cannot feedback and change the transport of the volcanic cloud caused by the heating inside the cloud due the absorption.

Figure 3: The difference between the solid and the dashed line is difficult to understand for the quick reader. My first impression was that the dashed line shows the total SO2 (volcanic plus background) and the solid the volcanic SO2 only.

Figure 3: Do you show an average over a specific region or the max values or another mean?

Line 194-195: I don't agree with this statement. As said before, in S21-3D the model misses the initial phase of the sulfur evolution and you get different results. A nudged model should simulate the transport well in case your timing and altitude of the injection is correct.

Line 197: Better first weeks.

Figure 4: Units? Do you show Tg (SO2) and Tg (SO4)? The same unit should be used, which is Tg (S).

Line 219: I don't understand the averaging of the 5th column. Do you show a global average? A zonal mean is an average over longitudes, so where is the difference to the other figures? Ahh, OK, the caption says NH average. Please, change the sentence and say NH average in the text as well.

Figure 5: Do you show the vertical integral, the burden here? Why do you subtract SO2 when you show SO4?

Line 230: This differs between the months. In July M16 seems to be the best.

Line 239: OK, but the altitude is substantial. I would say in the S21 simulations, especially S21-3D, the aerosol is at a too high altitude.

Line 249: Please, explain this with a few more words.

Line 257: Agree, but your aerosol in not at the right altitude. Timing of the eruption is important as well.

Figure 6: Please add the average area, e.g. zonal mean.

Fig 6e, are the colours correct? S21-3d show the smallest values and highest altitude.

Fig 7: global mean?

Line 270: All three simulation reproduce the observations. M16 has a larger bias, an error of roughly 25%. This is a bit, but it could be much worse.

Line 271 to 282: This last paragraph is right and important. However, your results are a bit more complex. I cannot agree that the S21 simulations are in general better than M16. S21-1D is better, but S21-3D simulates the volcanic cloud in a too high altitude. This impacts all results, especially lifetime and forcing.

Line 275: The limits are more complex. You do not mention microphysical processes. They are extremely important and highly non-linear. The sulfur evolution depends on your emission profile, emission rate, and on the timing of the eruption. It depends also on the microphysical scheme (Tilmes et al, 2023; Laakso et al 2022). You have to broaden your discussion and you should also discuss your results more critically.

**Specific comments:**

Laakso, Anton, Ulrike Niemeier, Daniele Visioni, Simone Tilmes and Harry Kokkola, Dependency of the impacts of geoengineering on the stratospheric sulfur injection strategy part 1: Intercomparison of modal and sectional aerosol module, Atmos. Chem. Phys., 22, 93–118, https://doi.org/10.5194/acp-22-93-2022, 2022

Tilmes, S., Mills, M. J., Zhu, Y., Bardeen, C. G., Vitt, F., Yu, P., Fillmore, D., Liu, X., Toon, B., and Deshler, T.: Description and performance of a sectional aerosol microphysical model in the Community Earth System Model (CESM2), Geosci. Model Dev., 16, 6087–6125, https://doi.org/10.5194/gmd-16-6087-2023, 2023.

---

## Author Comment (AC1)

Thank you for providing an interesting study.

I have a few questions here:

1. The title of the paper is featured in "high-resolution". Readers may expect that the differences in radiative forcing, etc., may be attributed to different resolutions. However, the major differences between the experiments M16 and S21-3D/S21-1D are not raised because S21-3D and S21-1D have a higher vertical resolution but because S21-3D and S21-1D assumed more sulfur was injected into the stratosphere. However, the title might make people think that S21-3D and S21-1D have the same integrated sulfur at the same vertical levels as M16, but only S21-3D and S21-1D have higher vertical resolutions. However, if we average S21-3D and S21-1D to a coarser vertical resolution than in M16, they still show more sulfur in the stratosphere than in M16.

   The difference in radiative forcing is mainly caused by the higher SO2 injection altitudes in our S21 dataset, not by the higher SO2 mass. The S21 simulations will show higher radiative forcing than M16 if implemented at coarser resolution, but that is not the main point here. The high-resolution dataset provides detailed information on the SO2 profile. The M16 dataset positioned most of the SO2 at too low altitudes. Hence, SO2 and the formed particulate SO4, is transported to the troposphere more rapidly in M16. Therefore, M16 underestimated both the magnitude and longevity of the volcanic impact on the climate. Please see further discussion on this in point 2 below.

2. In our study, we calculated about 58% of the SO2 (0.81 Tg) was injected directly into the stratosphere during the eruption (Wu et al., 2017, ACP). (There might be further troposphere-to-stratosphere exchange that may transport a little more sulfur into the stratosphere later.) The number (58%) is actually closer to the 75% from the M16 experiment in your study. 95% is too much, which is not in agreement with observations.

   Sandvik et al., (2021) compiled their high resolution SO2 data based on a UT/LS SO2 product provided by Fred Prata. Hence, SO2 located at low altitudes were not included in the data.

   Some differences in radiative forcing are caused by different SO2 masses in the simulations, but most of the differences stem from deeper stratospheric injections for the S21 simulations (see answer to point 1 above). The SO2 mass injected to the stratosphere differ by ~10% among the simulations, i.e. ~1.03 Tg for the S21 simulations (95% of 1.09 Tg) and 0.9 Tg (75% of 1.2 Tg) for M16. Decreasing the volcanic stratospheric SO2 mass in S21 by 10% over the entire vertical SO2 profile would not result in substantial differences in the simulated AODs.

3. Fig.7 further demonstrates the above problem. The AOD from experiments S21-3D and S21-1D looks to have better agreements with the AOD from CALIOP, which proves the results from S21-3D and S21-1D have significantly overestimated the AOD caused by the sulfate aerosol from the Sarychev eruption. Because the AOD from CALIOP is composed of all kinds of aerosol information (NOT only sulfate aerosol) unless you have excluded the other aerosol species from the CALIOP AOD.

CALIOP and the simulations both include background aerosol and volcanic SO2. We simulated the stratospheric AOD by adding SO2 to the stratosphere. The model includes also the background stratospheric aerosol. Sarychev added only small amounts of ash to the stratosphere. This is evident in the low depolarization ratios observed by CALIOP in the weeks following the eruption (e.g. Prata et al., 2017: https://doi.org/10.5194/acp-17-8599-2017), i.e. sulfate is the dominating component in the volcanic aerosol.

Regards,

Xue

---

## Author Comment (AC2)

The paper by Axebrink et al. highlights an important aspect for simulations of the sulfate aerosol evolution in the stratosphere after a volcanic eruption. They performed simulations with three different SO2 emission profiles and show that the results depend strongly on the profile used.

Emission rate, altitude, profile and timing are important for a realistic simulation. The evolution of the sulfate aerosol formation is very non-linear and depends on SO2 and background concentrations. The emission altitude determines the direction in which the volcanic cloud is transported in the model. Many of these aspects are nicely highlighted in the paper.

The paper is well written and reads well. I recommend publication after a few minor corrections.

Thank you for helping us to improve the manuscript. Your insights and comments on the microphysics clearly helped us to improve the interpretation of the simulations and their relation to the satellite observations.

Please see the answers to the reviewer's comments below.

**General:**

The article misses the aspect of microphysical processes and the consequent differences in particle size between the different simulations. Scattering of sulfate aerosols depends on the particle size (e.g. Laakso et al 2022). This could be another reason for the differences in the simulated forcing. My recommendation is to take a look at the particle size, e.g. calculate the effective radii, and include a short discussion.

We agree that the microphysical processes are of great importance for the climate impact of the stratospheric sulfate aerosol. We have added a figure of aerosol effective radii and one displaying the AOD divided by stratospheric sulfate mass to investigate the importance of the size distribution for the light reflection. In relation to this figure (new Figure 8), we have included a paragraph discussing the microphysics importance for aerosol light scattering and the differences between the simulations. See section 3.3.

The description of your experiments is a bit misleading. The table implies that your vertical grid of the S21s simulations has a vertical grid spacing of 200 m. This is not the case. Instead, you must interpolate the input data onto the model grid. Right? So it is not the resolution of the injection input data that is important. What is more important is the injection profile in the model. What does it look like and what does the model grid look like? Figure 6 shows quite similar profiles for June. So I assume that the height resolution of the input data disappears in the model. You can show and explain more details and show an initial phase profile.

We agree with the reviewer that Table 1 was misleading. We have changed one of the headings and the Table caption to clarify that the Table describes the input datasets. We have also added this sentence to the methods section: "When the SO$_2$ is emitted in the model it is interpolated to the model grid which is the same for all simulations.".

When the datasets are read in by the model, they are interpolated to the model grid and it is true that the effects of the resolution of the datasets are reduced. Since the emissions using the different datasets are emitted over several days and at different days it is not possible to produce one single injection profile. Nevertheless, the vertical SO2 profiles shown in Figure 3a are the total SO2 in the model 5 days after the eruptions started, i.e. the first day that the SO2 emissions in all simulations have been injected in the model and the first time at which the injections profiles from the simulations can be compared to each other. We have expanded the results section with regards to Figure 3a specifically focusing on the initial phase of the simulations.

A clear reason for the differences in the forcing calculations is the simulated altitude of the volcanic cloud (Fig 6). Surprising is the different behaviour between S21-1D and S21-3D. A better explanation should be added.

We have made more in-depth analysis of the differences between the simulations, including investigations of aerosol microphysics. Please the comments below.

**Specific comments:**

Line 38: What means significant here?

We have deleted that sentence to comply with remarks from reviewer #2.

Line 43: A residence time of several years is quite unrealistic.

We agree and deleted the word 'several'.

Line 49: No, the climate impact is estimated from the simulations. Satellites provide estimates on eruption rate and altitude.

We agree and changed the sentence to *"…Global modelers often use satellite-based observations of volcanic SO2 as input when simulating the volcanic impact on the stratosphere and climate…"*

Line 62: ….climate impact. Do you have a reference?

Our statement is based on pure logic. Differences in wind directions at different altitudes and locations may produce large differences in the spread and stratospheric lifetime of the volcanic sulfur. We changed the sentence to: *"… Small errors in horizontal or vertical transport may cause errors in the evolution of the $SO_2$ distribution (Tilmes et al. 2023) and transport of the formed sulfate particles, and ultimately in the resulting climate impact.…".*

Line 63: Maybe, but due to the non-linear nature of the aerosol formation your simulations miss the early phase of the eruption. The may lead to different particle sizes and aerosol concentration.

This is indeed an uncertainty and complication of our approach. However, we find only small differences in particle size distributions among the three simulations in the first week after the eruptions (SO2 injection to the model).

Line 72-73: Add a reference for the dataset.

We have added the reference to Mills et al. (2016).

2.2 Model description: Information on the vertical grid are important here. Table 1: Very misleading. Starting with 'Simulation name' and 'Vertical resolution' this could also be the vertical resolution of the model. What means single column? Should be in the model 0.95 x 1.25 as well.

Thank you for pointing this out. We agree that this was confusing. The table refers only to the input datasets and not the simulations. We have changed "Simulation name" to "Dataset name". We have also clarified this even more in the Table caption which now reads: "Properties for the three input SO$_2$ datasets."

Fig 1: b) shows the vertical integral, the burden, of SO2?

Yes, this is the vertically integrated SO2 in the S21-3D dataset. We have clarified this figure text also based on comments from Reviewer # 2. The figure caption now reads: *"…(a) Vertical SO$_2$ profiles for the three input datasets of each simulation. The vertical profile for M16 and S21-1D is the summed total injection for the eruption on the 15th and the 16th of June, whereas the vertical profile for S21-3D is the total injection on the 19th of June. (b) Vertically integrated total amount of SO$_2$ for the S21-3D dataset. The red triangle marks the location of the volcano Sarychev Peak. (c) latitudinally integrated total amount of SO$_2$ for the S21-3D input dataset. (d) longitudinally integrated total amount of SO$_2$ for the S21-3D input dataset…"*

Line 165: SO2 does not disappear.

We agree with the reviewer. We have rephrased this sentence.

Line 184 and line 187-188: This argument is valid if your volcanic cloud stays at the same altitude (in meters). But from Fig 6 you can see a different behavior of the cloud. So the sulfur evolution cannot be deduced from Fig 1S alone. Microphysical processes are also important. You may have different particle size, different sedimentation, and different altitude, all of which affect the lifetime.

We agree that the sulfur evolution cannot be deduced from Fig S1 alone. The relatively small differences in particle sizes among the simulations cannot explain the differences in residence times. The S21-3D simulation results in the largest particles and shows the longest duration of aerosol perturbation in the stratosphere. Hence, sedimentation cannot explain the differences between the simulations shown in our Figures (e.g. Figure 6 and 7).

Line 189: The wording is difficult here. In principle, a model simulates one out of many possible weather/climate conditions. So, a model has its own reality. As you nudge the model, the results should be close to reality. On the other hand, a strongly nudged model cannot feedback and change the transport of the volcanic cloud caused by the heating inside the cloud due the absorption.

We agree with the reviewer. Our intention is point out the importance of the transport. We have modified the sentence to clarify this.

Figure 3: The difference between the solid and the dashed line is difficult to understand for the quick reader. My first impression was that the dashed line shows the total SO2 (volcanic plus background) and the solid the volcanic SO2 only.

The solid line shows the volcanic SO2 in the stratosphere while the dashed line shows the total volcanic SO2 (stratospheric and tropospheric). The background has been removed by subtracting the noVolc SO2. We have changed the figure caption to make this clearer.

Figure 3: Do you show an average over a specific region or the max values or another mean?

These are globally summed values. We have added this information to the figure caption.

Line 194-195: I don't agree with this statement. As said before, in S21-3D the model misses the initial phase of the sulfur evolution and you get different results. A nudged model should simulate the transport well in case your timing and altitude of the injection is correct.

It is true that using the S21-3D dataset misses the sulfur evolution during the first 3-4 days. However, since the aerosol formation during those days occur under rather extreme conditions inside a concentrated volcanic cloud it is uncertain to what degree a global model can realistically simulate the aerosol formation during these days.

A nudged global model will simulate large scale features of air transport well but will still struggle to mimic mesoscale phenomena such as jet-streams due to the limited resolution of the model. This is of particular importance for volcanic eruptions that reach the midlatitude tropopause region. In the case of the Sarychev eruption, there was a rapid cross-Pacific transport of the volcanic $SO_2$ layers indicating that transport by jet-streams are important.

Line 197: Better first weeks.

We changed accordingly.

Figure 4: Units? Do you show Tg (SO2) and Tg (SO4)? The same unit should be used, which is Tg (S).

We have changed the figure according to the reviewer's suggestion. We have also updated the figure caption based on this.

Line 219: I don't understand the averaging of the 5th column. Do you show a global average? A zonal mean is an average over longitudes, so where is the difference to the other figures? Ahh, OK, the caption says NH average. Please, change the sentence and say NH average in the text as well.

*Thank you for pointing this out. We have corrected this according to the reviewer's suggestion.*

Figure 5: Do you show the vertical integral, the burden here? Why do you subtract SO2 when you show SO4?

*This was a typo. It should say that we have subtracted the SO4. This has been corrected.*

Line 230: This differs between the months. In July M16 seems to be the best.

*We agree with the reviewer on this and have modified the text to reflect this. The text now reads:*
*"Above these lowest altitudes, the model simulations have similar extinction coefficients as the CALIOP observations. During July, the M16 profiles bear most resemblance to the CALIOP profiles but after this month, the profiles from the S21 simulations have values more similar to the CALIOP observations."*

Line 239: OK, but the altitude is substantial. I would say in the S21 simulations, especially S21-3D, the aerosol is at a too high altitude.

*The (small) difference between the S21 simulations and CALIOP could be the result of the aerosol being placed at a too high altitude (due to aerosol formation or transport) but another explanation is slightly too high extinction values in the S21 simulations. During August to October the vertical profiles of the aerosol extinction coefficient have similar slopes for the S21 simulations and CALIOP at 14- 20km altitude (Figure 6). The simulations show slightly higher aerosol load (aerosol extinction) than the observations.*

*The difference between the S21 simulations and the satellite observations may (to a large degree) be explained by uncertainties in the injected SO2 mass. The S21 simulations overestimate the aerosol extinction coefficient values mostly in June-July. Although, the aerosol extinction coefficients differ between the S21 simulations and satellite observations, their difference is smaller than the typical uncertainty of volcanic SO2 estimates. In the case of the Sarychev eruptions the SO2 estimates ranges from 0.6 – 1.2 Tg. While the extinction coefficient values are non-linearly connected to the SO2 mass, our new figure (new Fig. 8) shows only small differences in the AOD response to SO2 (or rather AOD/SO4) among the three simulations. Hence, large part of the difference between the S21 simulations and the satellite observations likely stem from uncertainties in the SO2 estimates.*

Line 249: Please, explain this with a few more words.

*We have expanded this sentence.*

Line 257: Agree, but your aerosol in not at the right altitude. Timing of the eruption is important as well.

It could be that our aerosol is at a too high altitude. But the difference between the simulations highlights the importance of the vertical placement of the SO2 injections. We have changed the text to better reflect this and the sentence now reads: *"This large difference exemplifies the importance of the vertical placement of volcanic SO₂ injections in global climate models."*

Figure 6: Please add the average area, e.g. zonal mean.

We have added that these are zonal means.

Fig 6e, are the colours correct? S21-3d show the smallest values and highest altitude.

Yes, the colors are correct. Since CALIOP and the simulations have different tropopauses and latitudinal extent the averaging in this figure is not done over all the gridboxes used in the other figures. Also, since the S21-3D data is injected days later than the other datasets, this profile is averaged with more days without Sarychev aerosol present.

Fig 7: global mean?

Yes, thanks for pointing this out. We have added this information to the figure caption.

Line 270: All three simulation reproduce the observations. M16 has a larger bias, an error of roughly 25%. This is a bit, but it could be much worse.

We agree that the M16 simulations could reproduce the aerosol extinction coefficients in parts of the stratosphere during some months and decided to delete this statement (the last part of the sentence).

Line 271 to 282: This last paragraph is right and important. However, your results are a bit more complex. I cannot agree that the S21 simulations are in general better than M16. S21-1D is better, but S21-3D simulates the volcanic cloud in a too high altitude. This impacts all results, especially lifetime and forcing.

It is true that the S21-1D and S21-3D simulations predict aerosol load at higher altitudes than the observed aerosol extinction coefficient values. On the other hand, M16 underestimates the aerosol extinction coefficients at all altitudes during Aug-Oct.

Line 275: The limits are more complex. You do not mention microphysical processes. They are extremely important and highly non-linear. The sulfur evolution depends on your emission profile, emission rate, and on the timing of the eruption. It depends also on the microphysical scheme (Tilmes et al, 2023; Laakso et al 2022). You have to broaden your discussion and you should also discuss your results more critically.

We agree that the sulfur evolution is limited by more factors than SO2 vertical distributions alone. We have changed the sentence to clarify that the vertical SO2 profile is one of the factors influencing

the sulfur evolution (which in turn impacts the climate). The sentence now reads: *"These findings highlight the need to produce high-vertical resolution datasets of volcanic SO2 injections to the stratosphere and indicate that our present understanding of volcanic climate cooling is **in part** limited by the SO2 profiles."*

We have also added discussions regarding the microphysical processes to the manuscript. Please see our answers to comments above.

**Specific comments:**

Laakso, Anton, Ulrike Niemeier, Daniele Visioni, Simone Tilmes and Harry Kokkola, Dependency of the impacts of geoengineering on the stratospheric sulfur injection strategy part 1: Intercomparison of modal and sectional aerosol module, Atmos. Chem. Phys., 22, 93–118, https://doi.org/10.5194/acp-22-93-2022, 2022

Tilmes, S., Mills, M. J., Zhu, Y., Bardeen, C. G., Vitt, F., Yu, P., Fillmore, D., Liu, X., Toon, B., and Deshler, T.: Description and performance of a sectional aerosol microphysical model in the Community Earth System Model (CESM2), Geosci. Model Dev., 16, 6087–6125, https://doi.org/10.5194/gmd-16-6087-2023, 2023.

---

## Author Comment (AC3)

**Review of the manuscript "High-resolution stratospheric volcanic SO2 injections in WACCM", Axebrink et al**

Dear Editor, dear Authors,

The manuscript "High-resolution stratospheric volcanic SO2 injections in WACCM" by Axebrink et al discusses large-scale modelling of the Sarychev eruption in 2009, with the aim of investigating the importance of input injection parameters and other model set-up aspects in the description of stratospheric volcanic eruptions, towards the estimation of radiative impacts of such events. The topic of the manuscript is important and of certain interest for the ACP readers. There is an active ongoing scientific debate, at the international scale, about how to represent volcanic plumes (stratospheric but also tropospheric), and their impacts, with numerical modelling. The consistency of confined plumes (volcanic emissions, wildfires, etc) observations and modelling is still to be achieved, to be honest. For this reason, this manuscript has the potential to be an important contribution to this debate. Unfortunately, I have fundamental concerns about the model set-up and cannot recommend this manuscript for publication as it is, see Specific Comments 20 and 23-29. Based on these comments, I rather recommend clarification or re-design of the experiments before I can fully evaluate this manuscript. For this reason, for the moment, I have not evaluated the Results section, and I'm waiting for such clarifications before going further in this review. In addition, I have found the manuscript severely lacking in text quality (i.e. different statements without justification) and the literature citation (knowledge?) is also to be strongly improved, see Specific Comments 1-22.

Please find Specific Comments in the following. Please address all these comments and I will be happy to review a further manuscript version, if the Authors decide to resubmit it to ACP.

I am sorry if I cannot be more positive this time but I strongly encourage the Authors to address my comments, improve the manuscript, re-design and re-run the simulations if needed, and then resubmit a new manuscript version.

Regards.

We thank the reviewer for the valuable comments that helped us improve our manuscript. We have improved the description of the model simulations to make the model set-up clearer. We have also provided a more thorough explanation for our choices in the set-up of the model simulations and the implementations of the different datasets. See more details regarding this under our answers to Specific Comments 20 and 23-29. We have also improved the text quality in response to the reviewers' comments. For more detail see answers to Specific Comments 1-22.

Please see the answers to the reviewer's comments below.

**Specific Comments:**

1) L22-23: why a full stop between the two sentences?

We have changed this according to the reviewer's suggestion.

2) L23-24: "Aerosol emissions...greenhouse gases", please add one or more references for this statement.

We added a reference to Hansen et al. (2023).

3) L23: "These effects result in a net cooling...", not always! See the case of black carbon aerosols, e.g.: https://www.nature.com/articles/s41467-020-20482-9 or https://acp.copernicus.org/articles/22/9299/2022/ and others

Yes, in specific settings the aerosol effect can be warming or cooling, but this is not of relevance in our study of the global climate effect of volcanic SO2 injection to the stratosphere. We have no wish to go into details on wildfire emissions in this manuscript, which is covered in our previous work, e.g. Martinsson et al., 2022 and Friberg et al., 2023.

4) L25: "...natural sources", why only natural aerosol sources? What about anthropogenic aerosol sources?

We agree that both natural and anthropogenic sources are of importance. Here we decided to mention only the natural, due to the topic of the paper (volcanic SO2 injections to the stratosphere).

5) L28-29: Please rephrase: the SO2 emissions do not have a direct impact on the radiative balance, the subsequently formed sulphate aerosols have (it is said right after).

Yes, in its molecular form SO2 has very limited impact on the climate. Thank you for noticing this mistake in the phrasing. We have modified the sentence to clarify that we refer to climate effects of volcanic eruptions.

6) L29: the SO2 actually converts to binary solution droplets of sulphuric acid + water (e.g. https://agupubs.onlinelibrary.wiley.com/doi/full/10.1002/2015RG000511). Maybe this sentence can be rephrased accordingly.

Yes, it does form sulfuric acid which attracts water and form particulate matter. We have deleted the word 'aerosol' in the sentence.

7) L30: "...which can create years of...", please mention the very long lifetime of sulphate aerosols in the stratosphere

We agree that this should be mentioned. We have made changes according to the reviewer's suggestion.

8) L33: "explosiveness" --> "explosivity"

The reviewer is correct – explosivity is the correct word. We have changed accordingly.

9) L33: "the SO2 mass" --> "the mass of the injected SO2"

We have changed according to the reviewer's suggestion.

10) L35: please state clearly that the cooling of the Earth's climate system from volcanic eruption is *transient*

This information is now stated as: *"particulate matter, which can remain in the stratosphere for months or years inducing long-term negative radiative forcing by scattering incoming solar radiation"*.

11) L39-40: we are not at all in a background stratospheric aerosol condition, and the radiative effects of moderate stratospheric eruptions extends well beyond the "beginning of the 2000s": please mention, at least, more recent eruptions such as Raikoke 2019 (e.g. https://acp.copernicus.org/articles/21/535/2021/) and the very special case of the Hunga eruption 2022 (https://www.nature.com/articles/s43247-022-00618-z)

We agree – the present state is not a volcanically quiescent period. We decided to remove the sentence.

Our manuscript is focused on the June 2009 Sarychev eruptions. Other volcanic eruptions are covered in our previous work (e.g. Andersson et al., 2015, Friberg et al., 2018). We find post-Sarychev volcanic eruptions to be less relevant for our study, including Merapi, Nabro, Kelut, Calbuco, Ulawun, Manam, Ambae, Raikoke, Hunga Tonga-Hunga Ha'apai, etc.

12) L44-46: This sentence sounds like a repetition of what already said before and can be suppressed.

We wish to keep the sentence.

13) L44: "SPA, 2006" is rather "SPARC, 2006"? This looks like too generic as a reference and the Authors can easily find more specific references

We agree and now cite Deshler et al. (2008) and Robock et al. (2000) instead.

14) Please mention representative cases, like this one (plus others, in case): https://agupubs.onlinelibrary.wiley.com/doi/full/10.1029/2021JD035974

We do not understand in which sentence or paragraph the reviewer wishes us to add this reference.

15) L48-49: please briefly state how satellite observations are used by modelers. Synergies studies can be cited, e.g.: https://agupubs.onlinelibrary.wiley.com/doi/full/10.1029/2021JD035974 and https://acp.copernicus.org/articles/16/6841/2016/ and others

We understand that our sentence was misleading and changed it to *"Global modelers often use satellite-based observations of volcanic SO2 as input when simulating the volcanic impact on the stratosphere and climate".*

16) L49-50: "Most SO2..." "most" or "all"? How SO2 can be measured with active observations? Also, the question of the vertical resolution of satellite observations (especially in a nadir geometry) is complicated and should be briefly discussed here (e.g.: passive sensors do not lack vertical measurements but rather have limited vertical sensitivity, etc)

Satellite based SO2 sensors retrieve vertical information indirectly. We changed the sentence in line with the reviewer's suggestion.

17) L51-52: "*Clarisse et al. (2014) showed that IASI can provide SO2 data with vertical resolution down to ~2 km,*", this sounds a bit overestimated for infrared observations at the nadir, please check

This is the resolution that Clarisse et al. (2014) present in their paper. Please see the abstract of their paper.

18) L58: "column" --> "total vertical column"

The standard approach is to run ESMs with altitude resolved SO2 data as input, and not with single values of the height integrated SO2 mass (as the term *"total vertical columns"* suggest). The SO2 profile within the (vertical) column may however, in some simulations,

consist of uniform distributions, as well as for example triangular or Gaussian distributions.

As this sentence seem to have caused some confusion, we have rephrased it and the sentence now reads: *"ESM simulations of explosive volcanic eruptions' climate impact are generally run with vertical $SO_2$ profiles released above, or in the vicinity of, the volcano site (Timmreck et al. 2018)."*

19) L65: "*implemented*" --> the Authors mean "used as input"? See also L83

We have changed according to the reviewer's suggestion. We have removed the word implemented from the entire manuscript.

20) L73-76: I honestly did not understand the difference between second and third data sets. Please clarify.

*The third dataset has the same vertical profile as the second dataset but is released in single column (one lat x lon grid box only) rather than over several latitude and longitude grid-boxes. This is further explained in answers to the comments below.*

21) L79-81: this three-lines introduction can be suppressed as it is redundant

We wish to keep this as is.

22) L86: SO2 (AIRS) and aerosol (CALIOP) vertical profiles do not "have the same height profile" but the Authors assume this is the case, which implies the fact that the Authors assume that the SO2 and aerosol plumes are collocated. This has to be stated and the chemical/microphysical implications of this assumption should also be briefly mentioned.

Our sentence on L85-86 (*"The SO2 and aerosol observed from these instruments were assumed to be co-located and therefore have the same height profile"*) refers to the co-location of SO2 and aerosol, i.e. the assumption used to produce the high vertical resolution SO2 data in Sandvik et al. (2021). CALIOP's role in Sandvik et al. 2021, was to provide vertical information for the SO2 AIRS data.

The present manuscript, Axebrink et al. (2024), focuses on the simulations with data produced in Sandvik et al. (2021). Please see Sandvik et al. (2021) for further description on how SO2 data were compiled, i.e. explicitly how CALIOP and AIRS data were combined using FLEXPART.

23) L124: "*M16 is a single column (1D) emission dataset with a vertical resolution of 1 km.*" How can a "single column emission dataset" have a "vertical resolution of 1 km" <-- this means that the emissions are not based on a single column but on different vertical layers (at 1 km resolution).

With this sentence we would like to point out that the M16 dataset is released in one lat x lon grid box rather than in several lat x lon grid boxes around the volcano site. A column consists of different vertical layers that has a vertical resolution of 1 km.

The terminology 'single column' is commonly used within global modelling of volcanic eruptions e.g. Tilmes et al, 2023 which was recommended to us by reviewer #1.

24) L125: why M16's and S21-1D emissions are released at a different time interval (15-16/6) than    S21-3D (19/6)? If it is now known that Sarychev emissions were mainly on 19/6, why making simulations of the "wrong" days?

The Sarychev eruptions occurred mainly on the 15[th] and 16[th] of June. We have used the M16 dataset in WACCM in the same manner as in previous studies of this eruption with this model. The S21-3D dataset was released on the 19[th] June since the satellite dataset was recorded at that specific date. This is explained in Section 3.1.

25) L125-126: how SO2 is released during the 6-hours period? Is it a constant emission rate? Is there a peak at some time? Why only 12:00 to 18:00 for the two days and not before/after? This sound as an unphysical way of "erupting" for a volcano and should be fixed.

This is a standard approach for the WACCM modelling of Sarychev's eruption and in line with the methods in Mills et al., (2016). We use the same approach as previous studies to investigate the differences between the standard approach and our approach (different dataset). The emission rate is constant during these times.

26) L130-131: how SO2 is released here as well (cfr previous comment)? Same for the third dataset (L133-135)

The SO2 emission rate is constant during these hours.

27) L132-133: this is very puzzling. Do the Authors mixed-up the vertical and horizontal definition of "single column"? What's the actual meaning of "1D" here?

With "1D" we mean that the dataset is not spread over several latitude and longitude grid-boxes and is therefore not a 3D dataset. The single column dataset only has the vertical dimension and is therefore one dimensional. Se comment 23 with regards to the use of the term single column.

28) L151-152: using CALIOP data as a comparison data set is not completely satisfactory in terms of independence with the simulations, as one the simulation was partially initialised with CALIOP information (S21-3D)

We disagree. When compiling the high-resolution SO2 dataset, CALIOP data is only used to tell the vertical positions of the SO2 layers. No CALIOP aerosol data is used as an input to the model. It is of interest to investigate how well the model can simulate the transformation of SO2 into aerosol, the removal and transport of the aerosol in the model and the resulting AOD distributions over time. The high-resolution aerosol dataset from CALIOP is highly suitable to evaluate the latter. CALIOP's vertical resolution is more than a magnitude higher than that of other satellite borne stratospheric aerosol sensors.

29) Fig. 1 caption: In panel b, this is a daily average or at a specific hour? In panels c and d are also for S21-3D simulation? Please mention this in the caption.

This is the summed total emission. We have updated the figure caption to better describe this and to explain that figures c and d display the S21-3D dataset. The figure caption now reads: "(a) Vertical SO2 profiles for the three input datasets of each simulation. The vertical profile for M16 and S21-1D is the summed total injection for the eruption on the 15th and the 16th of June, whereas the vertical profile for S21-3D is the total injection on the 19th of June. (b) Vertically integrated total amount of SO2 for the S21-3D dataset. The red triangle marks the location of the volcano Sarychev Peak. (c) latitudinally integrated total amount of SO2 for the S21-3D input dataset. (d) longitudinally integrated total amount of SO2 for the S21-3D input dataset."

References:

Tilmes, S., Mills, M. J., Zhu, Y., Bardeen, C. G., Vitt, F., Yu, P., Fillmore, D., Liu, X., Toon, B., and Deshler, T.: Description and performance of a sectional aerosol microphysical model in the Community Earth System Model (CESM2), Geosci. Model Dev., 16, 6087–6125, https://doi.org/10.5194/gmd-16-6087-2023, 2023.

---

## Author Response (AR2)

Dear Authors,

I am now in possession of two reviews of your revised manuscript. While the first referee is satisfied with the changes to the original submission, the second is more critical and requests additional modifications. In line with their comments, I find that a few relatively minor but necessary changes are still required before the paper can be considered for publication.

Specifically, you may wish to consider the following suggestions:

1. Changing the paper title to emphasize the vertical distribution of SO2 rather than the resolution of the dataset, making it more consistent with the actual content and discussion in the paper. It would also be relevant to mention which eruption is under investigation (the June 2009 Sarychev eruption).

We agree that the title was a bit misleading. A major finding in our manuscript is the importance of compiling SO2 data with instruments that have high-vertical resolution and provide altitude information with high accuracy. Even though the CALIOP data has been available for ~15 years, we are likely the first group to simulate the eruptions as thin SO2 layers separated by km in altitude, resulting in a vertical profile with more modes than one. CALIOP, with its high accuracy in vertical profile estimates, is to our knowledge, the only instrument that could provide data of this kind. Our study is not only an investigation of the importance of the vertical distribution of the SO2. Such a sensitivity study could have been performed by many other groups using assumed SO2 profiles. A major importance of our study is that we use the first SO2 dataset that has been compiled with high accuracy vertical information and show that the model can perform more realistic estimates of the aerosol load when run with this data.

We have tried to stress this in our manuscript as well as in the new manuscript title: "Impact of SO2 injection profiles on simulated volcanic forcing for the Sarychev 2009 eruptions - investigating the importance of using high vertical resolution methods when compiling SO2 data".

We are aware that this title is longer than the original title. However, the simulations themselves are not the main achievement in this work. As stated above, a major importance of our study is that the simulations were performed with SO2 data that were compiled with high vertical accuracy and precision.

2. Adopting the suggestion (by Ref. 1 of the original manuscript) to show the injected SO2 profile in Figure 1 on the model vertical grid for June 15-16, as represented for S21-1D and M16. Including the location of the model levels would also be helpful for readers.

We have created a figure depicting the SO2 profiles during the first 6 days after the eruption (15-20 of June) to the supplementary. The figure includes the location of the model levels. We have also added a short paragraph discussing this figure in the results section.

3. Adjusting the statements discussing the need for high resolution to make them more consistent with the actual findings of the paper, which highlight the sensitivity to the vertical profile of SO2 (and the partitioning between troposphere and stratosphere) rather than the importance of fine

vertical resolution data. I am thinking, for instance, of the last sentence of the abstract or line 300 in the conclusion.

We agree. We have gone through the manuscript and changed the text to highlight that the vertical placement of the SO2 rather than the resolution of the dataset is the critical factor for the results. We have changed both the sentences suggested by the editor.

Please also address the specific and general comments by the referees.

We have answered all of the reviewers' comments.

Many thanks for the careful revision of the paper. I just have one tiny remark. Please add to Fig. 8 where you picked the radii. Are the plotted radii the maximum values or values at a certain altitude?

Thanks for this remark. The effective radius is the weighted geometrical mean for the entire stratosphere. This is stated in the caption for Figure 8.

After reviewing the authors' responses, I appreciate their efforts to address each question raised in the reviewers' comments and the community feedback, as well as the minor modifications made in response to the reviewers. However, I still have a lingering concern regarding the main focus of the study. Initially, I had hoped to see a demonstration of how the authors' SO2 emissions could be shown to be more accurate than those in M16, potentially due to improved resolution of both the volcanic SO2 plume height and mass. This would, ideally, result in better simulation outcomes compared to those based on M16. Unfortunately, this crucial point seems to remain unaddressed in the revised manuscript.

I acknowledge that the authors have undertaken simulations using their own developed volcanic SO2 emissions (S21-1D and S21-3D) instead of relying on the original volcanic SO2 inventory (M16), which is commendable and bold. However, the contribution of this work to our current understanding of volcanic impact simulation remains somewhat unclear to me.

While M16 is largely derived from satellite observations and may have a coarse resolution, it is generally not inaccurate when compared to observations. Although S21-1D and S21-3D offer high vertical resolution, the data must be interpolated to model levels, as pointed out by Reviewer #1. Additionally, as noted by other reviewers, S21-1D and S21-3D, particularly S21-3D, appear to overestimate SO2 mass and altitudes, leading to excessively high aerosol burdens and altitudes in the simulations. Consequently, the results from these simulations do not exhibit significant improvement over those based on M16. While S21-1D and S21-3D have been discussed in Sandvik et al. (2021), this does not inherently qualify them to enhance volcanic impact simulation performance. Even Sandvik et al. acknowledged the elevated and larger estimates compared to other studies. It seems that S21-1D and S21-3D may require further refinement before their application can be expanded.

M16 assumes uniform vertical distributions of SO2 after volcanic eruptions, which is a simplification that does not align with the observed distributions of aerosol after many volcanic eruptions.

Since SO2 sensors lack the high vertical information/resolution needed to retrieve precise SO2 vertical distributions, we use the formed aerosol particles as evidence for the vertical location and distribution of the injected SO2. CALIOP revealed that almost half of the aerosol formed from SO2 after Sarychev reached above the 380 K-isentrope, of which a fraction was transported to the tropics within the shallow Brewer-Dobson branch (e.g. Figure 8b in Friberg et al. 2018). Hence, half of the SO2 was injected above the 380 K-ienstrope. This is not the only example when M16 misrepresents the vertical distribution of the SO2. For example, the Kasatochi eruptions in Aug 2008 injected SO2 to varying altitudes forming two main aerosol layers separated by ~5 km in altitude (Figure 2 in Andersson et al., 2015; Friberg et al., 2018). M16 treats Kasatochi as a uniform layer spanning 10-18 km altitude, whereas CALIOP shows that the aerosol was positioned in two layers centered at ~10 and 16 km in the extratropics (the vertical position is better described by isentropes), with most of the aerosol located close to the tropopause. Hence, M16 does not capture the complex vertical distribution of Sarychev 2009 and Kasatochi 2008.

Regarding the manuscript's title, it may inadvertently be misleading. The emphasis on "high resolution" could suggest more than what the current findings substantiate, and the title might not accurately reflect the core results of the study, potentially leading to skepticism.

We have changed the title of the manuscript to align with this comment.

The methodology used to derive S21-1D and S21-3D (Sandvik, 2021, AMT) might benefit from additional consideration. The approach of constructing profiles by tracking volcanic aerosol signals from CALIOP above the tropopause might contribute to missing data at lower altitudes and the tendency to place $SO_2$ injections in the stratosphere.

This would cause bias if using $SO_2$ data that captures $SO_2$ in the entire atmospheric column. We therefore used an UTLS $SO_2$ product produced by Fred Prata and used CALIOP backscattering data to tell the vertical $SO_2$ profile. Hence, low altitude $SO_2$ data does not impact the $SO_2$ profile. CALIOP provided precise altitude resolved data at high resolution.

These comments are offered with the hope of enhancing the scientific significance of the manuscript to live up to the standard of ACP. I commend the authors for their work and encourage them to continue refining their approach.

PS. I noticed that in the replies, the author answered that the results could be compared with CALIOP aerosol data because CALIOP is independent of their data. However, it's important to note that the initial $SO_2$ profile was derived by tracing the CALIOP aerosol data backward. This raises a question about the extent to which the CALIOP data can be considered independent in this context.

CALIOP was used to estimate the altitude of the $SO_2$ measured by AIRS, while the mass estimate where produced by AIRS data. Hence, CALIOP was not used to determine the amount of $SO_2$ put into the model. The $SO_2$ is put into the model as a gas phase compound. The model then simulates aerosol formation, transport of both $SO_2$ and aerosol, the evolution of the size distribution of aerosols in relation to the injected $SO_2$. The simulated aerosol extinction coefficients computed months after the eruption depend on the injected $SO_2$ profiles but also on many other parameters. Hence it is realistic to evaluate the model performance against CALIOP measurements during the months after the eruptions. We are interested in simulating the time evolution of the aerosol formation and transport in the stratosphere. Using CALIOP to retrieve the vertical $SO_2$ profile generates initial $SO_2$ profiles, but the subsequent aerosol formation and atmospheric dynamics is independent of CALIOP. Hence, we argue that CALIOP is well-suited for the comparison used in our manuscript.